# Electrochemical Fingerprint Analyses Using Voltammetry and Liquid Chromatography Coupled with Multivariate Analyses for the Discrimination of Schisandra Fruits

**DOI:** 10.3390/molecules30010048

**Published:** 2024-12-26

**Authors:** Koichi Machida, Akira Kotani, Tomoya Osaki, Ayaka Kobayashi, Kazuhiro Yamamoto, Hideki Hakamata

**Affiliations:** Department of Analytical Chemistry, School of Pharmacy, Tokyo University of Pharmacy and Life Sciences, Hachioji 192-0392, Japan; kmachida@toyaku.ac.jp (K.M.); y161038@toyaku.ac.jp (T.O.); y184088@toyaku.ac.jp (A.K.); yamamoto@toyaku.ac.jp (K.Y.); hakaman@toyaku.ac.jp (H.H.)

**Keywords:** Schisandrae Sphenantherae Fructus, Schisandrae Chinensis Fructus, Schisandra fruit, voltammetry, HPLC, multivariate analysis

## Abstract

The appearances of Schisandrae Sphenantherae Fructus (SSF) and Schisandrae Chinensis Fructus (SCF) are very similar. Thus, being able to distinguish between SSF and SCF is useful for the quality control of these herbal medicines. In this study, two kinds of electrochemical fingerprint analyses using voltammetry or HPLC with electrochemical detection (HPLC-ECD) were developed in combination with multivariate analysis for discriminating between SSF and SCF. The data sets of the oxidation current values from voltammograms of SSF and SCF samples ranging from +0.5 to +1.0 V were applied to perform a partial least squares discriminant analysis (PLS-DA). Moreover, the data sets of the current heights of the characteristic target peaks on the chromatograms at an applied potential of +0.9 V were also applied to perform PLS-DA. In each PLS-DA obtained from the voltammograms and chromatograms, the scores for the SSF samples were plotted on a different region compared with the scores for the SCF samples. Considering the results of the cross-validation, the HPLC-ECD clearly discriminated between the SSF and SCF samples when compared with the voltammetry. In conclusion, our results show that the present electrochemical fingerprint analyses coupled with PLS-DA are useful as a means for discriminating between the SSF and SCF samples.

## 1. Introduction

Schisandrae Sphenantherae Fructus (SSF) and Schisandrae Chinensis Fructus (SCF) are widely used in traditional Chinese medicines and are officially recorded as two different kinds of crude drugs in the Chinese Pharmacopoeia [1,2]. SSF is the dried ripe fruit of *Schisandra sphenanthera* Rehd. et Wils., whereas SCF is that of *S. chinensis* (Turcz.) Baill., indicating the original plant source of SSF is different from that of SCF [1]. In the comparison of SSF and SCF, there are differences in the constituents and their contents such as lignans, organic acids, triterpenoids, flavonoids, and polysaccharides [3,4,5,6,7], and, thus, there are also differences in the diseases treated by SSF and SCF. For instance, SSF is more suitable for the treatment of chronic coughs, mild breathlessness, and skin inflammation [8], while SCF has better cardiovascular-protective, neuroprotective, and tonic activities [9,10,11].

As mentioned above, SSF and SCF have different pharmaceutical and chemical properties; however, their very similar appearances and tastes make it difficult to distinguish them [12]. Moreover, because the price of SCF, which has a higher content of bioactive lignans, is more expensive when compared with that of SSF, it has been reported that SCF has been intentionally mixed with SSF and has been sold on the market [5,13,14]. SCF is listed under the name Schisandra Fruit in the Japanese Pharmacopoeia (JP) and as Northern Schisandra Fruit in the United States Pharmacopeia (USP), whereas SSF is not listed in either pharmacopeia [15,16]. Thus, to be able to discriminate between SSF and SCF by instrumental analysis would be useful to maintain a stable supply and the quality control of these Schisandra fruits. To achieve these purposes, various instrumental analyses using high-performance liquid chromatography with UV detection (HPLC-UV), gas chromatography with mass spectrometry (GC-MS), LC with mass spectrometry (LC-MS/MS), and supercritical fluid chromatography with diode array detection (SFC-DAD) have been proposed to determine lignans and/or examine the fingerprint profiles of the constituents in SSF and SCF samples by several groups including us [12,17,18,19,20,21,22,23].

Electrochemical analysis is an attractive and useful technique for specifically determining electro-active compounds, and it has various merits for its relatively low cost, low power consumption, eco-friendliness, and ability for rapid measurements [24,25,26,27]. It has also been reported that lignans, which have methoxy groups on the aromatic rings, are electro-active compounds [28]. If there are qualitative and/or quantitative differences in the electrochemical behaviors of the electro-active compounds derived from SSF and SCF, it should be possible to develop an electrochemical analysis to distinguish between these fruits. To the best of our knowledge, an electrochemical analysis such as voltammetry and electrochemical detection in an HPLC (HPLC-ECD) system including fingerprint analysis has not been attempted for the analysis of the electro-active compounds found in SSF and SCF.

In this study, two kinds of electrochemical fingerprint analyses using voltammetry or HPLC-ECD were developed in combination with multivariate analysis for discriminating between SSF and SCF.

## 2. Results and Discussion

### 2.1. Voltammetric Behaviors of the Methanol Extracts of Schisandra Fruits

To record a voltammogram, a potential sweep was started at 0 V vs. Ag/AgCl to the positive direction, and the sweep was reversed at +1.0 V vs. Ag/AgCl. Figure 1 shows typical voltammograms obtained from a methanol extract of SSF, SCF, and Schisandra fruit of JP-grade (JPSF) samples. On the voltammogram of the SSF sample, defined and smooth oxidation waves were observed at around +0.75 V and +0.90 V, respectively, as shown in Figure 1a. No reduction waves were observed when the potential sweep was performed in the negative direction, indicating that the obtained oxidation products during the positive scan were not reduced during the negative scan. Thus, the electrochemical oxidation reactions of the electro-active compounds derived from the SSF sample were found to be irreversible (Figure 1a). On the voltammogram of the SCF sample, a smooth oxidation wave was observed at around +0.90 V and was also irreversible (Figure 1b). In order to confirm that Schisandra fruits distributed in Japan correspond to SCF, JPSF samples were used in the present study. The voltammetric behavior of the JPSF sample was nearly the same as that of the SCF sample (Figure 1c). On the voltammograms of the SCF and JPSF samples, a defined oxidation wave at around +0.75 V was not observed as in the voltammogram of the SSF sample.

To determine the amount of Schisandra fruit sample required to prepare a methanol extract, the relationships of the oxidation current values at +0.90 V and amounts of SSF, SCF, and JPSF samples were examined by voltammetry. The oxidation current values showed a linear relationship with the amounts of the SSF, SCF, and JPSF samples to be dissolved in 10 mL of methanol, ranging from 0.125 g to 1.50 g (*r* > 0.992). Considering the linear ranges, 0.75 g of the Schisandra fruit samples were used for preparing a methanol extract in the present voltammetry.

In the voltammetry for the SSF samples of the 14 lots from the seven Chinese provinces, the shapes of the voltammograms were similar to one another. In the cases of the voltammetry for the SCF and JPSF samples, these results were also similar to one another. It was found that the voltammetric behaviors for the SSF samples differed from those for the SCF and JPSF samples.

### 2.2. PCA and PLS-DA Using Voltammograms of Schisandra Fruits

In this study, a principal component analysis (PCA) and partial least squares discriminant analysis (PLS-DA) were performed to demonstrate a means to discriminate between SSF and SCF using reliable and objective values derived from statistical methods. Oxidation current values of 250 points were recorded during the potential sweep, ranging from +0.5 V to +1.0 V with a scan rate of 0.02 V/s and a sampling interval of 0.1 s/point (0.5/0.02/0.1 = 250). The PCA for discriminating between the SSF and SCF samples was applied to the data sets of the oxidation current values of 250 points at each potential. The data sets of the oxidation current values obtained from the first scanned voltammograms were used to avoid lowering the accuracy and precision by electrolytic product accumulations on the working electrode surface, which occurred by multiple cycles of the potential sweep. The oxidation current values at around +0.5 V were similar for the SSF, SCF, and JPSF samples, while the oxidation currents at around +1.0 V for the SSF samples were larger than those for the SCF and JPSF samples. In the PCA and PLS-DA, the data sets of the oxidation current values without the normalizing would cause a problem of discriminating only when the oxidation current values are large, at around +1.0 V. To avoid the above-mentioned problem, normalizing, which is commonly used in multivariate analysis, was applied, and thus the characteristic oxidation waves were recognized on the voltammograms of the SSF and SCF samples.

In the present PCA, the variances of the first, second, third, fourth, and fifth principal components (PC1, PC2, PC3, PC4, and PC5) were 54.6, 36.4, 5.7, 2.8, and 0.3%, respectively. Thus, the cumulative variance of PC1 and PC2 was 91.0%, indicating that these PCs were applicable for identifying the inhomogeneity of the samples used in the present PCA. Figure 2 shows the PCA score plots obtained from the data sets of the oxidation current values on the voltammograms ranging from +0.5 to +1.0 V. The plots for the PC1 and PC2 scores of the SSF samples were clustered in the first and fourth quadrants, while those of the SCF and JPSF samples were clustered in the second and third quadrants.

To discriminate between the SSF and SCF samples more clearly, a PLS-DA was performed using the same data sets as for the PCA. In the PLS-DA, the JPSF samples were grouped as the SCF samples. Figure 3 shows the PLS-DA score plots between components 1 and 2. As shown in Figure 3, the scores for the SSF samples were plotted on a different region compared with the scores for the SCF and JPSF samples, indicating that the SSF samples were identified as Schisandra fruits that are different from the SCF and JPSF samples. It was also found that the present voltammetry coupled with PLS-DA can be applied to discriminate between SSF and SCF samples and recognize the equality between the SCF and JPSF samples. An interesting point of the present study is that the data sets, which were obtained from the voltammograms including a smooth oxidation wave, as shown in Figure 1, were applicable to the discrimination of the Schisandra fruit samples. In the present voltammetry analysis, the data sets of the oxidation current values on the voltammograms for PLS-DA were provided within 2 min, and thus, the present voltammetry allows for a multitude of analyses to distinguish between the SSF and SCF samples.

Next, cross-validation was performed to evaluate the performance of the classifier, which is represented by accuracy, the coefficient of determination (*R*^2^), and the predictive coefficient of determination (*Q*^2^). Cross-validation was conducted by varying the number of the components used in the PLS-DA from 1 to 5, and the results of accuracy, *R*^2^, and *Q*^2^ are shown in Appendix A. The *Q*^2^ values for components 1, 2, 3, 4, and 5 were 0.86878, 0.86779, 0.86578, 0.86721, and 0.89626, respectively. The closer the *Q*^2^ value is to 1 means an unknown Schisandra fruit sample can be correctly discriminated as an SSF or SCF sample by the classifier in the PLS-DA. These cross-validation results show that the classifier in the PLS-DA performs sufficiently to discriminate between SSF and SCF samples.

### 2.3. Chromatographic Behaviors of the Methanol Extracts of Schisandra Fruits

As shown in Figure 1, the voltammetric behaviors for the SSF samples differed from those for the SCF and JPSF samples. Based on the separations of the electro-active compounds achieved by HPLC-ECD, we hypothesize that HPLC-ECD can not only provide more accurate discrimination for Schisandra fruit samples compared to voltammetry but can also help identify key compounds that contribute to the discrimination. Figure 4a shows a typical chromatogram obtained from a methanol extract of SSF, and many peaks derived from the electro-active components in the methanol extract of SSF are observed on the chromatogram. In the present study, the chromatographic peaks, which have a signal-to-noise ratio (S/N) of more than 10, were analyzed as target peaks by a multivariate analysis. On the chromatograms obtained from the methanol extract of the SSF samples, the target peaks, which appeared at 6.5, 9.2, 11.0, 13.4, 21.1, 26.2, 31.9, 43.0, 70.1, and 90.5 min, were analyzed, and these peaks are called I, II, III, IV, VI, VII, VIII, X, XII, and XIII, respectively.

Figure 4b,c show typical chromatograms obtained from the methanol extracts of the SCF and JPSF samples, respectively. The peaks III, IV, VI, VII, VIII, IX, X, and XIII, which appear on the chromatogram of the SSF sample, were also observed on the chromatograms obtained from the ethanol extracts of the SCF and JPSF samples. Moreover, other target peaks appeared at 17.6 and 38.5 min, and these peaks are called V and XI, respectively. In the chromatogram measurements for the other SSF, SCF, and JPSF samples, the peaks I, II, and XII were absent from the chromatograms of the methanol extracts from the SCF and JPSF samples, and the peaks V and XI were not in those of the SSF samples.

Hydrodynamic voltammograms of the electro-active compounds in the methanol extracts of the Schisandra fruit samples were measured to determine the optimal applied potential in the HPLC-ECD (Appendix A). The peak heights of the electro-active components at +1.0 V were highest except for peak XIII; however, the baseline noises increased in comparison with those at +0.9 V. Considering the S/N of each target peak, +0.9 V was selected as the applied potential in the present HPLC-ECD system.

### 2.4. PCA and PLS-DA Using the Chromatograms of Schisandra Fruits

The current peak heights I–XIII observed on the chromatograms from the methanol extracts of the Schisandra fruit samples were used as data sets to perform the PCA and the PLS-DA. In the present PCA, the variances of the first, second, third, fourth, and fifth principal components (PC1, PC2, PC3, PC4, and PC5) were 63.6, 14.8, 12.2, 4.0, and 2.1%, respectively. Thus, the cumulative variance of PC1 and PC2 was 78.4%, indicating that these PCs were also applicable to identifying the inhomogeneity of the samples used in the present PCA. Figure 5a shows the PCA score plots obtained from the data sets of the current peak heights I–XIII. The plots for the PC1 and PC2 scores of the SSF samples were clustered in the first and fourth quadrants, while those of the SCF and JPSF samples were clustered in the second and third quadrants. These results in Figure 5a show that the SSF samples were distinguishable from the SCF and JPSF samples, while the SCF and JPSF samples were recognized as being the same.

Figure 5b shows the PCA loading plots for the PC1 and PC2 of the Schisandra fruit samples. The loading plots for the PC1 of peaks I, II, XII, and XIII show positive PC1 values, while those of peaks IV, V, IX, and XI show negative PC1 values. Therefore, it was found that the former peaks were important factors in identifying the SSF samples, and the latter peaks were important factors for the SCF and JPSF samples.

Similarly to the voltammetry, a PLS-DA was performed using the same data sets of the current peak heights I–XIII. In the PLS-DA, the JPSF samples were grouped as the SCF samples. Figure 6 shows the PLS-DA score plots and loading plots between components 1 and 2. As shown in Figure 6a, the scores for the SSF samples were plotted in a significantly different region when compared with the scores for the SCF and JPSF samples, indicating that the SSF samples were clearly identified as Schisandra fruits that are different from the SCF and JPSF samples. Moreover, the PLS-DA loading plots in Figure 6b showed that important target peaks were identified to discriminate between the SSF and SCF samples, similarly to Figure 5b. It was found that the present HPLC-ECD coupled with PLS-DA is a useful strategy to discriminate between SSF and SCF samples and recognize the equality between SCF and JPSF samples. Using the HPLC-ECD, the regions of the scores for the SSF samples were remarkably separated from those for the SCF and JPSF samples.

Cross-validation was performed for the PLS-DA, and the performance results are shown in Appendix A. The *Q*^2^ values were 0.95123, 0.95429, 0.94966, 0.94909, and 0.94663 for components 1, 2, 3, 4, and 5, respectively. The *Q*^2^ values in the PLS-DA by HPLC-ECD were larger than those derived from voltammetry, indicating that the performance of the classifier in the PLS-DA using HPLC-ECD was superior to that using voltammetry. Thus, the discriminations made between the SSF and SCF samples by the HPLC-ECD were significantly clear when compared with those made by the voltammetry. Although the HPLC-ECD requires measurement times of 100 min for one analysis, the PCA and PLS-DA using HPLC-ECD can provide accurate discrimination for the Schisandra fruit samples when compared with that using the voltammetry. Moreover, another advantage of the HPLC-ECD is that the Schisandra fruit samples can be discriminated without the use of expensive reference standards of natural compounds.

## 3. Materials and Methods

### 3.1. Reagents

All chemicals and solvents were reagent-grade. Methanol and formic acid were purchased from FUJIFILM Wako Pure Chemicals (Osaka, Japan). Pure water was prepared using an ultrapure water system (RFU666HA, ADVANTEC, Tokyo, Japan).

### 3.2. Schisandra Fruit Samples

Schisandra fruit samples used in this study are listed in Appendix A. The SSF samples were gathered from the Hunan (two lots), Henan (five lots), Jiangxi (three lots), Guangxi (one lot), Shanxi (one lot), Ningxia (one lot), and Guangdong (one lot) provinces of China. Meanwhile, the SCF samples were gathered from the Liaoning (nine lots), Hebei (four lots), and Jilin (four lots) provinces of China. The JPSF samples (two lots), which were produced in the Liaoning province of China, were purchased from Tochimoto Tenkaido (Osaka, Japan). Other JPSF samples (one lot), which were produced in the Liaoning province of China, were purchased from Uchida Wakanyaku (Tokyo, Japan).

### 3.3. Voltammetry

#### 3.3.1. Apparatus and Electrodes

The voltammograms were obtained from a computer-controlled electrochemical system (HZ-3000, Hokuto Denko, Tokyo, Japan) and recorded at a scan rate of 0.02 V/s. The sampling interval was set at 0.1 s/point. Plastic-formed carbon (PFC, 3 mm diameter disk, Tsukuba Materials Information Laboratory, Ltd., Ibaraki, Japan), Ag/AgCl, and platinum wire were used as the working, reference, and auxiliary electrodes, respectively.

#### 3.3.2. Preparation of a Test Solution for the Voltammetry

A pulverized Schisandra fruit sample (0.75 g) was dissolved in 10 mL of methanol, and the extraction into methanol was performed by ultrasonication for 20 min at room temperature. This solution was centrifuged for 5 min at 3000 r.p.m., and then the supernatant was filtered through a membrane (pore size 0.45 μm, Chromatodisc 4A, Kurabo, Osaka, Japan). A test solution was prepared by mixing the filtrate (1 mL) and methanol–water–formic acid (60:40:1, *v*/*v*/*v*) containing 40 mM KCl (3 mL). The test solution was poured into the beaker-type electrochemical cell to perform the voltammetry.

### 3.4. HPLC-ECD

#### 3.4.1. HPLC-ECD System and Conditions

An HPLC-ECD system was equipped with a vacuum degasser (BG-34, Flom, Tokyo, Japan), a pump (AI-12, Flom), an auto sample injector (L-2200, Hitachi, Tokyo, Japan), an octadecasilyl-silica (ODS) column (Capcell Pak C18 UG120, 150 × 1.0 mm, i.d., 3 μm, Osaka Soda, Osaka, Japan), a column oven (CTO-10AS, Shimadzu, Kyoto, Japan), and an electrochemical detector (LC-4C, BAS, Tokyo, Japan). In the electrochemical cell (Radial flow cell, BAS), glassy carbon, Ag/AgCl, and stainless steel were used as the working, reference, and auxiliary electrodes, respectively.

A mobile phase of a methanol–water–formic acid mixture (60:40:1, *v*/*v*/*v*) was flowed at 30 μL/min. The ODS column was maintained at 40 °C. The applied potential was set at +0.9 V vs. Ag/AgCl.

#### 3.4.2. Preparation of a Test Solution for the HPLC-ECD Analysis

A pulverized Schisandra fruit sample (0.25 g) was dissolved in 20 mL of methanol, and the extraction into methanol was performed by ultrasonication for 20 min at room temperature. This solution was centrifuged for 5 min at 3000 r.p.m., and then this solution was filtered through a membrane (pore size 0.45 μm, Chromatodisc 4A). The filtrate (0.1 mL) was diluted to 10 mL with a mobile phase and 5 μL of the diluted solution was injected into the HPLC-ECD system.

### 3.5. Principal Component Analysis and Partial Least Squares Discriminant Analysis

MetaboAnalyst 6.0 [29] was utilized to perform PCA and PLS-DA. MetaboAnalyst is an integrated platform for metabolomics data analysis, offering the capability to perform a multivariate analysis based on common statistical strategies and/or machine learning techniques. On the platform, the data sets of the oxidation current values obtained by voltammetry were processed with normalizing and autoscaling, while the data sets of the current heights of the characteristic target peaks obtained by HPLC-ECD were processed with autoscaling. Normalizing and autoscaling are useful to perform multivariate analysis [30] and their processes and effectiveness are explained in the Appendix A, including Appendix A. These processed data sets were used to perform the PCA and PLS-DA.

## 4. Conclusions

Two kinds of electrochemical fingerprint analyses using voltammetry or HPLC-ECD were developed for the discrimination between the SSF and SCF samples. In these analyses, the data sets of oxidation current values from the voltammograms or the current heights of characteristic target peaks from the chromatograms were applied to perform the PCA and PLS-DA. From the results of the PCA and PLS-DA, it was confirmed that the SSF samples were different from the SCF samples and the JPSF samples were equivalent to the SCF samples. Our study is the first to show that voltammetry and HPLC-ECD can be applied as electrochemical fingerprint analyses for discriminating between SSF and SCF samples based on multivariate analysis. Thus, the electrochemical fingerprint analyses coupled with PCA or PLS-DA are practical and useful for the stable supply and quality control of Schisandra fruit.

## Figures and Tables

**Figure 1 molecules-30-00048-f001:**
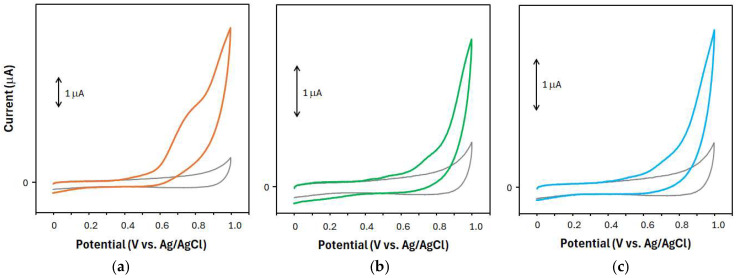
Cyclic voltammograms obtained from the methanol extracts of (**a**) SSF (orange line), (**b**) SCF (green line), and (**c**) JPSF (blue line) samples. The methanol extracts were prepared by dissolving pulverized Schisandra fruit in methanol at a concentration of 0.075 mg/mL. The methanol extract of each Schisandra fruit sample (1 mL) was mixed with methanol–water–formic acid (60:40:1, *v*/*v*/*v*) containing 40 mM KCl (3 mL). This solution was used to measure a voltammogram. When a background voltammogram, which is shown as a solid gray line in each figure, was measured, methanol was mixed instead of the methanol extract of each Schisandra fruit sample. The scan rate was set at 0.02 V/s.

**Figure 2 molecules-30-00048-f002:**
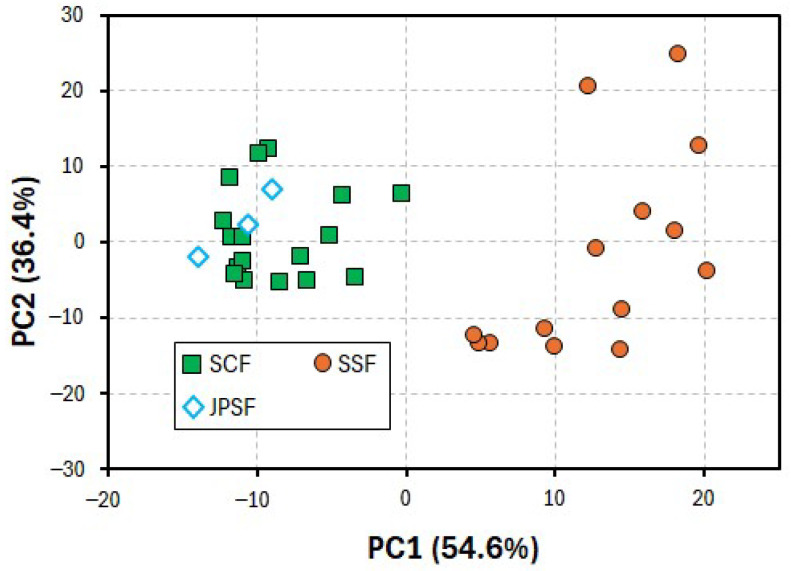
PCA plots obtained from data sets of oxidation current values on voltammograms ranging from +0.5 to +1.0 V vs. Ag/AgCl.

**Figure 3 molecules-30-00048-f003:**
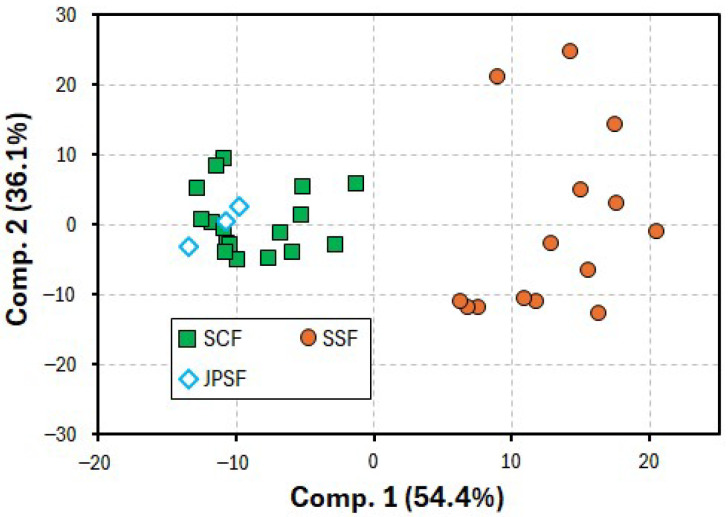
PLS-DA score plots obtained from data sets of oxidation current values on voltammograms ranging from +0.5 to +1.0 V vs. Ag/AgCl.

**Figure 4 molecules-30-00048-f004:**
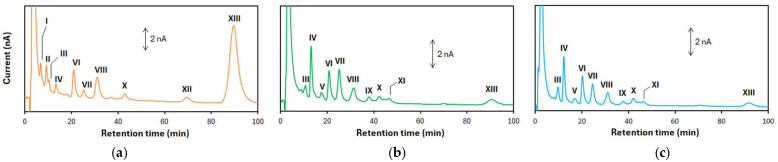
Chromatograms of methanol extracts of (**a**) SSF, (**b**) SCF, and (**c**) JPSF samples.

**Figure 5 molecules-30-00048-f005:**
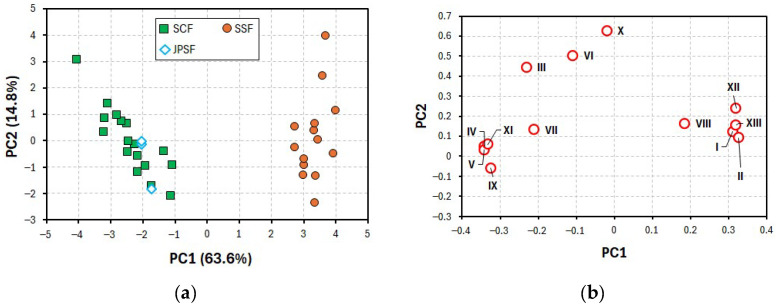
PCA (**a**) score plots and (**b**) loading plots from data sets of current peak heights I–XIII on the chromatograms.

**Figure 6 molecules-30-00048-f006:**
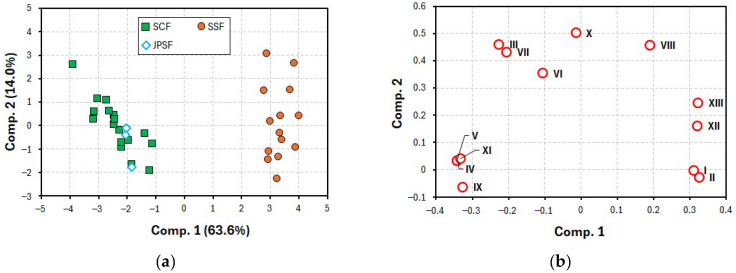
PLS-DA (**a**) score plots and (**b**) loading plots from data sets of current peak heights I–XIII on the chromatograms.

## Data Availability

The original contributions presented in the study are included in the article; further inquiries can be directed to the corresponding authors.

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
