# Peer review of "Electrochemical Fingerprint Analyses Using Voltammetry and Liquid Chromatography Coupled with Multivariate Analyses for the Discrimination of Schisandra Fruits"

_molecules, 2024, doi:10.3390/molecules30010048_

Round 1
Reviewer 1 Report (New Reviewer)
Comments and Suggestions for Authors
This article used the electrochemical fingerprint analysis together with HPLC to achieve the discrimination of SSF and SCF. However, there are some issues to be addressed.
1. From line 72-79, it described the oxidation peaks of SSF, SCF and JPSF. However, some of the peaks are hard to observe in Figure 1, such as the peak at 0.9V for SSF and 0.9V for SCF. Please give some explanation or show more clear detail of the curves.
2. It shows that the PLS and PLS-DA analysis is able to discriminate SCF and SSF, then what is the meaning to use HPLC? In fact, the HPLC curves of SCF and SSF are already different. Besides, the electrochemical curves of SCF and SSF are also different. What is the meaning to use PLS or PLS-DA method?
3. The aim of this article is to discriminate SSF and SCF, the what is the meaning of using JPSF?
4. There is no supplementary file supported.
Comments on the Quality of English LanguageThe expression needs to be improved.
Author Response
Comment:
- From line 72-79, it described the oxidation peaks of SSF, SCF and JPSF. However, some of the peaks are hard to observe in Figure 1, such as the peak at 0.9V for SSF and 0.9V for SCF. Please give some explanation or show more clear detail of the curves.
Reply: Thank you for your valuable suggestion. The voltammograms in Figure 1 have been vertically expanded to show clearer detail of the curves. As shown in each voltammogram of Figure 1, a smooth oxidation wave with changed behaviors observed at around +0.9 V.
- It shows that the PLS and PLS-DA analysis is able to discriminate SCF and SSF, then what is the meaning to use HPLC? In fact, the HPLC curves of SCF and SSF are already different. Besides, the electrochemical curves of SCF and SSF are also different. What is the meaning to use PLS or PLS-DA method?
Reply: Thank you for your valuable comments. The PCA and PLS-DA were performed to discriminate between SCF and SSF, not only based on voltammetric and/or chromatographic behaviors but also using reliable and objective values derived from statistical methods. It was found that the voltammetric behaviors for the SSF samples differed from those for the SCF and JPSF samples. Based on the separations of the electro-active compounds achieved by HPLC-ECD, we hypothesize that HPLC-ECD can not only provide more accurate discrimination for Schisandra fruit samples compared to the voltammetry but also help identify key compounds that contribute to the discrimination. To address the points mentioned above, the following sentence has been added to the revised manuscript.
Page 3, line 113: In this study, principal component analysis (PCA) and partial least squares discriminant analysis (PLS-DA) were performed to demonstrate a means to discriminate between SSF and SCF using reliable and objective values derived from statistical methods.
Page 5, line 177: As shown in Figure 1, the voltammetric behaviors for the SSF samples differed from those for the SCF and JPSF samples. Based on the separations of the electro-active compounds achieved by HPLC-ECD, we hypothesize that HPLC-ECD can not only provide more accurate discrimination for Schisandra fruit samples compared to the voltammetry but also help identify key compounds that contribute to the discrimination.
- The aim of this article is to discriminate SSF and SCF, the what is the meaning of using JPSF?
Reply: Thank you for your comments. To clarify the use of JPSF, the following sentence has been added to the revised manuscript.
Page 2, line 83: In order to confirm that Schisandra fruits distributed in Japan correspond to SCF, JPSF samples were used in the present study.
- There is no supplementary file supported.
Reply: Apologies for the oversight. Figure S1 was not cited in the main text of the original manuscript. In the revised manuscript, the original Figure S1 has been renumbered and cited as Figure S2 in the main text. This change was made because, in the revised manuscript, the original Figure S1 (page 9, line 318) was cited after the original Figure S2 (page 5, line 201). To ensure sequential numbering, we have swapped the figure numbers accordingly.
Comments on the Quality of English Language: The expression needs to be improved.
Reply: The revised paper has been proofread by Prof. Eric. M. Skier, a native speaker of American English and a professor of English at the School of Pharmacy, Nihon University, Japan.

Reviewer 2 Report (New Reviewer)
Comments and Suggestions for Authors
The manuscript of ID: molecules-3313740 entitled “Electrochemical fingerprint analyses using voltammetry and liquid chromatography coupled with multivariate analyses for the discrimination of Schisandra fruits” reports on the discrimination of Schisandra fruits two kinds of electrochemical fingerprint analyses using voltammetry or HPLC with electrochemical detection (HPLC-ECD). In addition, a multivariable analysis for discriminating SSF and SCF was used. This work has several strengths, however, I have some questions and suggestions that arose when I reviewed the submitted article. I hope they are useful for improving the final version of the work.
Remarks:
1. The abbreviation "JPSF" should be defined and explained in the initial instance of its use.
2. Section 2.1.: „No reduction waves were observed when the potential was swept to the negative direction, and thus it was found that these were irreversible (Figure 1A).” – it should be specified what was irreversible
3. Caption to Fig. 1: The concentration of the components is not given. It seems reasonable to posit that this is because the extract was provided and the concentration is unknown. Nevertheless, to facilitate comprehension for the reader, this information should be provided. What was the supporting electrolyte and potential sweep rate? These data should be added to this caption.
4. Did the authors attempt to use a different supporting electrolyte for voltammetric studies than that containing formic acid and KCl. At a potential of about +1.0 V Cl- ions undergo oxidation, distorting the image.
5. Is it not worth using a boron-dopped diamond electrode as a working electrode in the presented studies that is characterized by a low background current at highly positive potential values?
6. How background voltammogram was obtained? What supporting electrolyte was used?
In conclusion, I recommend a revision of the manuscript.
Author Response
Comment:
- The abbreviation "JPSF" should be defined and explained in the initial instance of its use.
Reply: Apologies for the oversight. The abbreviation “JPSF” has been defined as follows:
Page 2, line 73: Figure 1 shows typical voltammograms obtained from a methanol extract of SSF, SCF, and Schisandra fruit of JP grade (JPSF) samples.
- Section 2.1.: “No reduction waves were observed when the potential was swept to the negative direction, and thus it was found that these were irreversible (Figure 1A).” – it should be specified what was irreversible.
Reply: Thank you for your comments. To provide more detail, the following sentence has been added to the revised manuscript.
Page 2, line 77: No reduction waves were observed when the potential sweep was performed to the negative direction, indicating that the obtained oxidation products during the positive scan were not reduced during the negative scan. Thus, the electrochemical oxidation reactions of the electro-active compounds derived from the SSF sample were found to be irreversible (Figure 1A).
- Caption to Fig. 1: The concentration of the components is not given. It seems reasonable to posit that this is because the extract was provided and the concentration is unknown. Nevertheless, to facilitate comprehension for the reader, this information should be provided. What was the supporting electrolyte and potential sweep rate? These data should be added to this caption.
Reply: Thank you for your valuable suggestions. The caption for Fig. 1 has been revised as follows:
Figure 1. Cyclic voltammograms obtained from the methanol extracts of (a) SSF, (b) SCF, and (c) JPSF samples.
The methanol extracts were prepared by dissolving pulverized Schisandra fruit in methanol at a concentration of 0.075 mg/mL. The methanol extract of each Schisandra fruit sample (1 mL) was mixed with a methanol-water-formic acid (60:40:1, v/v/v) containing 40 mM KCl (3 mL). This solution was used to measure a voltammogram. When a background voltammogram, which is shown as a solid gray line in each figure, was measured, methanol was mixed instead of the methanol extract of each Schisandra fruit sample. The scan rate was set at 0.02 V/s.
- Did the authors attempt to use a different supporting electrolyte for voltammetric studies than that containing formic acid and KCl. At a potential of about +1.0 V Cl- ions undergo oxidation, distorting the image.
Reply: Thank you for your comments. Different supporting electrolytes, such as LiClO₄ and phosphoric acid, were successfully used to perform the present electrochemical fingerprint analyses. In this study, KCl was selected as the supporting electrolyte because it is commonly used in aqueous media. Additionally, formic acid was chosen because the present conditions are intended to be applied in an LC-MS/MS analysis in future studies.
As noted in the revised caption for Fig. 1, a supporting electrolyte containing KCl was used to measure the background voltammograms. No oxidation wave was observed, as shown in Figure 1. Therefore, under the present voltammetric conditions used for the electrochemical fingerprint analysis, the oxidation of Cl⁻ ions can be considered negligible.
- Is it not worth using a boron-dopped diamond electrode as a working electrode in the presented studies that is characterized by a low background current at highly positive potential values?
Reply: Thank you for your valuable suggestions. As you mentioned, the characteristics of a boron-dopped diamond (BDD) electrode would be worth in the present study. However, to the best of our knowledge, commercially available BDD disk electrodes for voltammetry are not available, and we lack the resources to fabricate BDD electrode as other researchers might. While BDD films that is used for wastewater treatment are available on the market, they are not intended for voltammetry and are prohibitively expensive. Although your suggestion is highly interesting and relevant, it is challenging to immediately incorporate the use of BDD electrodes into the present study.
- How background voltammogram was obtained? What supporting electrolyte was used?
Reply: Regarding the background voltammogram, it was measured using a mixture of methanol (1 mL) and a methanol-water-formic acid (60:40:1, v/v/v) containing 40 mM KCl (3 mL). Therefore, KCl and formic acid were included as supporting electrolytes. This information has been clarified in the caption to Figure 1, as addressed in response to comment No. 3.

Round 2
Reviewer 2 Report (New Reviewer)
Comments and Suggestions for Authors
The authors have addressed all the submitted comments.
I have no further remarks at this time.
I recommend accepting the paper for publication.
This manuscript is a resubmission of an earlier submission. The following is a list of the peer review reports and author responses from that submission.
Round 1
Reviewer 1 Report
Comments and Suggestions for Authors
Here two kinds of electrochemical fingerprint analyses using voltammetry or HPLC-ECD were developed in combination with a multivariable analysis for discriminating SSF and SCF. The data sets of oxidation current values on voltammograms of SSF and SCF samples ranges in good region of 0.5 - 1.0 The HPLC-ECD clearly discriminated between the SSF and SCF samples when compared with the voltammetry. Finally, the presented electrochemical fingerprint analyses coupled with PLS-DA were useful for the discrimination between SSF and SCF samples. The manuscript raised some questions in methodolgies and results, while the other parts looks normally acceptable and of interest for the specific readers. Therefore, I recommend reconsideration major revision.
The authors need to present at least 5 cycles of voltage sweep to find accurate Oxidation potential and present it in paper.
Very difficult to speak about reproducibility of the oxidation peaks in CVs of such extracts. Therefore a DPV method could be helpful.
Authors are suggested to present one DPV to reach more accurate conclusion about oxidation potentials of the supernatant solutions.
Methanol is generally not a good solvent for Electrochemical studies due to self-oxidation in shorter positive sweep be it on Glassy carbon or Platinum as Working electrode. Therefore, the use of 60% Methanol in preparing test samples are not recommended. Can the authors present CV of neat Methanol under identical experimental conditions for the supernatants.
Recent literature spanning novel use of electrochemical methods should be cited:
1. Adv. Sci. 2024, 11, 2306635.
2. The Journal of Physical Chemistry C 124 (38), 20974-20980
Author Response
The Response to Reviewer Comments
Reviewer #1:
Thank you very much for your careful review and valuable comments on our manuscript. Those comments are valuable and very helpful for improving our paper, as well as the important guiding significance to our researches. I am sending herewith the revised manuscript and our response to reviewers.
The authors need to present at least 5 cycles of voltage sweep to find accurate Oxidation potential and present it in paper.
Reply: Thank you for your valuable suggestions. In our voltammetry, current values are more significant factors than potential as the data sets for the multivariate analyses. As the reviewer’s comment, the accurate potential of the oxidation peak might be found by 5 cycles of the potential sweep. In our voltammetry, electrolytic products, which can be easily removed by rinsing the electrode using methanol, would be accumulated during multiple cycles of potential sweep on the working electrode surface, fearing the poor precision of current values. Thus, data sets obtained from the first scanned voltammograms were used to apply accurate and precise current values in multivariate analyses. The following sentences were added to explain the above contents.
Page 3, line 108: The data sets of the oxidation current values obtained from the first scanned voltammograms were used to avoid lowering accuracy and precision by electrolytic product accumulations on the working electrode surface, which occurred by multiple cycles of the potential sweep.
Very difficult to speak about reproducibility of the oxidation peaks in CVs of such extracts. Therefore a DPV method could be helpful.
Reply: As reviewer’s suggestions, oxidation waves obtained by DPV would be clearer than those by cyclic voltammetry. The measurements of cleared oxidation waves by DPV would be progressed quantitative analyses for electro-active components in SSF and SCF samples, and the research project including extended purposes from this study would like to examine in further works. The following sentences were added to explain the above contents.
Page 4, line 141: Although differential pulse voltammetry (DPV) would provide more defined oxidation waves compared with the cyclic voltammetry, the data sets of the oxidation current values on the voltammograms were sufficiently achieved to discriminate SSF and SCF samples by PCA and PLS-DA.
Authors are suggested to present one DPV to reach more accurate conclusion about oxidation potentials of the supernatant solutions.
Reply: An interesting point of the present study is the data sets, which are obtained from the voltammograms including smooth oxidation wave as shown in Figure 1, not DPV, are applicable to the discrimination of Schisandra fruit samples. The following sentences were added to explain the above contents.
Page 4, line 145: An interesting point of the present study is the data sets, which are obtained from the voltammograms including smooth oxidation wave as shown in Figure 1, not DPV, are applicable to the discrimination of Schisandra fruit samples.
Methanol is generally not a good solvent for Electrochemical studies due to self-oxidation in shorter positive sweep be it on Glassy carbon or Platinum as Working electrode. Therefore, the use of 60% Methanol in preparing test samples are not recommended. Can the authors present CV of neat Methanol under identical experimental conditions for the supernatants.
Reply: Thank you very much for your comments. As the reviewer pointed out, there are many papers regarding methanol oxidation, for example, a methanol fuel cell [a]. However, a mixture of methanol and water is commonly used as a solvent for voltammetry and HPLC with electrochemical detection [b-d]. As shown in Figure 1, no methanol oxidation waves are observed on the cyclic voltammograms of the background. Thus, the effects of methanol oxidation are minor under our voltammetric conditions.
[a] D.Y. Chung, Methanol electro-oxidation on the Pt surface: revisiting the cyclic voltammetry interpretation, J. Phy. Chem. C, 2016, 120, 9028-9035.
https://doi.org/10.1021/acs.jpcc.5b12
[b] R. Jain, et al., Voltammetric determination of cefixime in pharmaceuticals and biological fluids, Anal. Biochem., 2010, 407, 79-88.
https://doi.org/10.1016/j.ab.2010.07.027
[c] J.P. Hart, et al., Voltammetric behaviour of phylloquinone (vitamin K1) at a glassy-carbon electrode and determination of the vitamin in plasma using high-performance liquid chromatography with electrochemical detection, analyst, 1984, 109, 477-481. https://doi.org/10.1039/AN9840900477
[d] J.M.P. Carrazon, et al., Electroanalytical study of sulphamerazine at a glassy-carbon electrode and its determination in pharmaceutical preparations by HPLC with amperometric detection, Talanta, 1992, 39, 631-635. https://doi.org/10.1016/0039-9140(92)80072-L
Recent literature spanning novel use of electrochemical methods should be cited:
- Adv. Sci. 2024, 11, 2306635.
- The Journal of Physical Chemistry C 124 (38), 20974-20980
Reply: Thank you for your suggestions. However, K. Tieriekhov, et al. reports a magnetoelectrochemical rotation using bipolar electrochemistry and the magnetically induced Lorentz force [1], while F. Tassinari, et al., performs enantioselective chiral separation using spin-polarized electrodes [2]. Because our methods are based on amperometry such as voltammetry and HPLC with electrochemical detection, the citations of the suggested papers do not seem to enhance our manuscript. We reached this idea due to the description regarding references from the editor as follows: If the reviewer(s) recommended references, critically analyze them to ensure that their inclusion would enhance your manuscript. If you believe these references are unnecessary, you should not include them. I am sorry, but I would like to decline the reviewer’s suggestions.
[1] K. Tieriekhov, et al., Wireless magnetoelectrochemical induction of rotational motion, Adv. Sci. 2024, 11, 2306635.
https://doi.org/10.1002/advs.202306635
[2] F. Tassinari, et al., Spin-dependent enantioselective electropolymerization, J. Phys. Chem. C., 2020, 124, 20974−20980.
https://dx.doi.org/10.1021/acs.jpcc.0c06238

Reviewer 2 Report
Comments and Suggestions for Authors
The paper entitled “Electrochemical fingerprint analyses using voltammetry and liquid chromatography coupled with multivariate analyses for the discrimination of Schisandra fruits” proposes an innovative method to distinguish between Schisandrae Sphenanthera, and Schisandrae Chinensis Fructus, two different herbal medicines in the Chinese Pharmacopoeia with a different application in traditional Chinese medicine. These two fruits have different pharmaceutical and chemical properties, but their very similar appearance and taste make it difficult to distinguish between them. The Schisandrae Chinensis Fructus is also in the Japanese and United States of America pharmacopoeias. In this work, the oxidation currents in the voltammograms and the output of HPLC with electrochemical detection (HPLC-ECD) resorting to PCA and partial least squares discriminant analysis (PLS-DA) were evaluated for discriminating the two fruits. The cross-validation results showed that the HPLC-ECD discriminated between the Schisandrae Sphenanthera and Schisandrae Chinensis Fructus samples.
The proposed work is original and highly deserving of interest. The results indicate that HPLC-ECD has significant potential for developing robust PLS-DA models for discriminating Schisandrae fruit. The paper is well-organized, with an adequate introduction section, and the results are properly presented and discussed. However, the construction and validation of the models could be further developed to enhance the paper. Specifically:
1. Variable selection: This point, has not been thoroughly explored in the manuscript. Data in Figures 1, 4, and S2 only give an overview of raw data.
2. Model validation: One of the risks of PLS-DA is overfitting. While there are several strategies to avoid this, it is always advisable to test the model's robustness using a set of validation samples that were not involved in its construction (external validation). Was there any external validation set used?
Author Response
The Response to Reviewer Comments
Reviewer #2:
Thank you very much for your careful review and valuable comments on our manuscript. Those comments are valuable and very helpful for improving our paper, as well as the important guiding significance to our researches. I am sending herewith the revised manuscript and our response to reviewers.
The paper is well-organized, with an adequate introduction section, and the results are properly presented and discussed. However, the construction and validation of the models could be further developed to enhance the paper. Specifically:
Reply: We fully acknowledge your concern about the risk of overfitting in supervised learning methods such as PLS-DA. We would like to address your point regarding the variable selection and the model validation in comments 1 and 2, respectively.
- Variable selection: This point, has not been thoroughly explored in the manuscript. Data in Figures 1, 4, and S2 only give an overview of raw data.
Reply: Thank you very much for your valuable comments. For the reviewer, the data sets of the oxidation current values and the current peak heights for multivariate analyses were attached as an Excel file. To avoid confusion due to huge data sets, typical voltammograms and chromatograms were shown in Figures 1 and 4, respectively. The hydrodynamic voltammograms of Figure S2 are just measured to select the optimal applied potential in the HPLC-ECD, and thus we think that it is sufficient to show the results.
- Model validation: One of the risks of PLS-DA is overfitting. While there are several strategies to avoid this, it is always advisable to test the model's robustness using a set of validation samples that were not involved in its construction (external validation). Was there any external validation set used?
Reply: We recognize the critical importance of proper validation in assessing model performance and preventing overfitting. In our study, we implemented a comprehensive cross-validation strategy that inherently includes external validation:
We applied k-fold cross-validation, specifically using 5-fold cross-validation, as described in the Supplementary Information (SI) of our manuscript. This method effectively creates multiple external validation sets throughout the process, as each fold serves as an independent test set for the model built on the remaining folds. The cross-validation results in Tables S1 and S2 (SI) show that we obtained appropriate results without overfitting. This approach ensures that each sample in our data sets serves as both training and validation data across multiple iterations, providing a robust assessment of the model's performance on "unknown" data. To clarify our validation procedure and address your concern more explicitly, we have added the following explanations in the revised SI. In the revised manuscript, we will ensure that the description of our validation strategy is more prominently featured and thoroughly explained. This will provide readers with a comprehensive understanding of the steps taken to ensure the robustness and reliability of our PLS-DA model.
The footnotes in Tables S1 and S2: For model validation, we used 5-fold cross-validation. This method inherently involves external validation, as each fold serves as an independent test set during the process. This approach ensures that our model is tested on "unknown" data throughout the validation process, allowing for a rigorous evaluation of its generalizability and robustness to overfitting.

Round 2
Reviewer 1 Report
Comments and Suggestions for Authors
The authors did not perform any of the experimental studies suggested during revision stage while accepting that those suggestions are helpful. Therefore, I can not unambiguously accept the conclusion and the hypothesis of this research without any of the suggested data. So, I recommend reject.
Reviewer 2 Report
Comments and Suggestions for Authors
The authors' responses were enlightening.
Using k fold cross validation during training is suitable. However, In this cross-validation, the data is divided into k sets. Then, for each fold, a model is trained using k-1of the folds as training data and the resulting model is validated on the remaining part of the data. This process is repeated for the K validation sets and the final result is an average of the results obtained for all sets. Thus, all samples end up participating in the construction of the model, integrating the validation series 1 time and being part of the training series (k- times).There are other approaches, but generally follow the same principles. For this reason, to make an independent assessment of the model's capacity, it is desirable to have an independent test set with samples that do not participate in this k fold cross validation process. Therefore I kindly ask the authors to consider revising the senstence “This approach ensures that our model is tested on "unknown" data throughout the validation process, allowing for a rigorous evaluation of its generalizability and robustness to overfitting” in the footnote on table S1.
Given the authors' choices, the predictions made in this paper probably are optimistic.